# Primary Driver Mutations in *GTF2I* Specific to the Development of Thymomas

**DOI:** 10.3390/cancers12082032

**Published:** 2020-07-24

**Authors:** Rumi Higuchi, Taichiro Goto, Yosuke Hirotsu, Yujiro Yokoyama, Takahiro Nakagomi, Sotaro Otake, Kenji Amemiya, Toshio Oyama, Hitoshi Mochizuki, Masao Omata

**Affiliations:** 1Lung Cancer and Respiratory Disease Center, Yamanashi Central Hospital, Yamanashi 400-8506, Japan; lumi.hgc.236@gmail.com (R.H.); dooogooodooo@me.com (Y.Y.); nakagomi.takahiro@gmail.com (T.N.); sotaro.otake@gmail.com (S.O.); 2Genome Analysis Center, Yamanashi Central Hospital, Yamanashi 400-8506, Japan; hirotsu-bdyu@ych.pref.yamanashi.jp (Y.H.); amemiya-bdcd@ych.pref.yamanashi.jp (K.A.); h-mochiduki2a@ych.pref.yamanashi.jp (H.M.); m-omata0901@ych.pref.yamanashi.jp (M.O.); 3Department of Pathology, Yamanashi Central Hospital, Yamanashi 400-8506, Japan; t-oyama@ych.pref.yamanashi.jp; 4Department of Gastroenterology, The University of Tokyo Hospital, Tokyo 113-8655, Japan

**Keywords:** thymoma, driver mutation, sequencing, molecular barcoding

## Abstract

Thymomas are rare mediastinal tumors that are difficult to treat and pose a major public health concern. Identifying mutations in target genes is vital for the development of novel therapeutic strategies. Type A thymomas possess a missense mutation in *GTF2I* (chromosome 7 c.74146970T>A) with high frequency. However, the molecular pathways underlying the tumorigenesis of other thymomas remain to be elucidated. We aimed to detect this missense mutation in *GTF2I* in other thymoma subtypes (types B). This study involved 22 patients who underwent surgery for thymomas between January 2014 and August 2019. We isolated tumor cells from formalin-fixed paraffin-embedded tissues from the primary lesions using laser-capture microdissection. Subsequently, we performed targeted sequencing to detect mutant *GTF2I* coupled with molecular barcoding. We used PyClone analysis to determine the fraction of tumor cells harboring mutant *GTF2I*. We detected the missense mutation (chromosome 7 c.74146970T>A) in *GTF2I* in 14 thymomas among the 22 samples (64%). This mutation was harbored in many type B thymomas as well as type A and AB thymomas. The allele fraction for the tumors containing the mutations was variable, primarily owing to the coexistence of normal lymphocytes in the tumors, especially in type B thymomas. PyClone analysis revealed a high cellular prevalence of mutant *GTF2I* in tumor cells. Mutant *GTF2I* was not detected in other carcinomas (lung, gastric, colorectal, or hepatocellular carcinoma) or lymphomas. In conclusion, the majority of thymomas harbor mutations in *GTF2I* that can be potentially used as a novel therapeutic target in patients with thymomas.

## 1. Introduction

Thymoma is a relatively rare mediastinal tumor that is difficult to treat [1,2]. Based on the histological classification by the World Health Organization, thymomas can be categorized into the types A, AB, B1, B2, and B3 depending on the tumor cell morphology and proportion of coexisting lymphocytes [3]. Thymomas of the A category are the least aggressive with the best prognosis; the extent of aggressiveness increases and the prognosis worsens according the order: type A, AB, B1, B2, and B3 [4,5]. Owing to the absence of effective treatment other than surgical resection, there is an urgent need to develop novel drug therapies for patients with inoperable advanced-stage thymomas and those with postoperative relapses of the tumor [6,7,8,9].

Analyzing the mutant genes present in thymomas is important in identifying novel treatment strategies. Recent studies showed a missense mutation (chromosome 7 c.74146970T>A) in *GTF2I* (GTF: general transcription factor) present with high frequency in type A thymomas [10,11]. Thymomas are encapsulated tumors. Type AB thymomas histologically comprise a complex mixture of type A and B thymomas. Thus, it seemed unreasonable to hypothesize that mutations in *GTF2I* account for the development of the type A component, with other mechanisms responsible for the development of the type B component. Thus, we focused on the importance of mutations in *GTF2I* in the development of type B thymomas using targeted sequencing coupled with techniques in molecular barcoding: more sensitive and specific assays than the whole-exome sequencing approach used in previous studies [10,11]. We expect that candidates that are commonly mutated in the majority of thymomas will help develop novel therapeutic targets in molecular targeting and gene therapies in the future.

## 2. Results

### 2.1. Patient Characteristics

We analyzed samples from 22 patients with thymomas who had undergone surgery (*n* = 21) or surgical biopsy (*n* = 1) at Yamanashi Central Hospital between January 2014 and August 2019. Table 1 shows the clinicopathologic characteristics of the patients, such as the age, sex, histology, tumor size, stage, smoking status, and diagnosis of myasthenia gravis. Among the 22 patients, 12 and 10 were males and females, respectively, and 14 and 8 were smokers and non-smokers, respectively. Using histological examination, there were five, three, seven, five, and two patients with type A, AB, B1, B2, and B3 tumors (Table 1). There were no cases of micronodular thymoma. The 22 patients recruited in this study were divided according to the Masaoka stages: stage I (*n* = 7), II (*n* = 12), III (*n* = 2), and IV (*n* = 1). The maximum tumor diameter ranged from 20 mm to 95 mm (mean tumor diameter, 43.6 ± 22.8 mm). The age of the patients ranged between 42 and 81 years (66.5 ± 12.6 years). One patient with type B2 thymoma exhibited comorbidity with myasthenia gravis (Case 16; Table 2).

### 2.2. Targeted Sequencing

Table 2 shows the data obtained from the sequencing. The sequencing coverage ranged between 1623–9140 (mean ± SD: 3566 ± 2309). We detected point mutations in *GTF2I* in all the type A and AB thymomas; several type B thymomas were also positive for these *GTF2I* mutations. The type A and B portions of the type AB thymomas harbored mutant *GTF2I*. The allele fraction with the mutant *GTF2I* was lower in type B thymomas compared to in type A thymomas; this could be attributed to the presence of normal cells in the tumor specimens. Mutations in *GTF2I* were detected in 14 out of 22 patients with thymomas (64%). Mutant *GTF2I* was detected in at least one sample from all the subtypes of thymomas (A, AB, B1, B2, and B3). Thus, the *GTF2I* mutation may well be called a prevalent mutation in thymomas in general.

### 2.3. PyClone Analysis

In our analysis of somatic mutations, there was a need to alleviate the allelic imbalances due to normal-cell contamination, especially in lymphocyte-rich type B thymomas. In this context, PyClone analysis was performed to estimate the cellular frequency patterns of mutations in a population of tumor cells. Mutant *GTF2I* was harbored in ~20%–90% of the tumor cells among all the thymomas (Figure 1), suggesting a high cellular prevalence of mutant *GTF2I*. This *GTF2I* mutation appeared to trigger clonal expansion and is retained ubiquitously within the tumors of the same clone.

### 2.4. PD-L1 Expression

PD-L1 expression was evaluated immunohistochemically in thymomas with the *GTF2I* mutation (GTF2I+) and without (*GTF2I*−; Figure 2A,B). Samples from the *GTF2I*+ group included 3 PD-L1-positive and 11 negative cases, whereas those in the *GTF2I*− group comprised 7 PD-L1-positive and 1 negative case. The distribution of positive and negative cases was significantly different between the two groups (*p* < 0.05; Chi-square test). The staining intensity of PD-L1 was significantly higher in the *GTF2I*+ group compared to in the *GTF2I*− group (Figure 2C), which suggests the mutually exclusive presence of PD-L1 expression and the *GTF2I* mutation in tumor cells.

### 2.5. Correlation between the Clinical Factors and Genomic Profiles of Patients

The age, sex, smoking habits, tumor size, histological type, Masaoka stage, and PD-L1 expression were assessed using multivariate analysis to identify factors affecting the mutation status of *GTF2I*. Based on a Cox proportional hazards model, the histology and PD-L1 expression were factors that determined the presence of mutations in *GTF2I*; sex (*p* = 0.41), age (*p* = 0.43), smoking habit (*p* = 0.68), tumor size (*p* = 0.97), and tumor stage (*p* = 0.46) did not correlate with the mutation status of *GTF2I*. In essence, mutant *GTF2I* was detected at a higher extent in type A and AB thymomas compared to in types B1–B3. PD-L1-negative thymomas harbored mutant *GTF2I* significantly more frequently when compared to PD-L1-positive thymomas (HR: 12.60, 95% CI: 1.19–133.89).

### 2.6. Peripheral Blood Parameters

Peripheral blood markers were examined to better characterize the cases in reference to the clinical benefits in immunotherapy. An elevated lactate dehydrogenase (LDH) level was reported to be indicator of tumor burden that is typically associated with lower response rates to immunotherapy, while an elevated platelet-lymphocyte ratio (PLR) was also associated with lower response rates in patients treated with immunotherapy [12]. The pre-surgery blood exam data were analyzed, which revealed that serum LDH levels were significantly higher in thymomas with mutant *GTF2I* than those with wildtype *GTF2I* (thymomas with mutant *GTF2I*, 224.3 ± 13.3; thymomas with wildtype *GTF2I*, 183.7 ± 12.1; *p* < 0.05). In addition, the PLR was significantly higher in thymomas with mutant *GTF2I* than those with wildtype *GTF2I* (thymomas with mutant *GTF2I*, 137.2 ± 13.1; thymomas with wildtype *GTF2I*, 97.0 ± 15.7; *p* < 0.05).

### 2.7. Specificity of GTF2I Mutations in Thymoma

In order to examine the specificity of the *GTF2I* mutation, patients with other malignant diseases were also enrolled in the study. The patients who underwent surgery or biopsy at our hospital between January 2014 and August 2019 were enrolled without bias, and they exhibited a wide range of histology and stages. Mutant *GTF2I* was not detected in other carcinomas, such as brain cancer, lung cancer, gastric cancer, colorectal cancer, hepatocellular carcinoma, and breast cancer, or in lymphomas (*n* = 20 for each, Table 3), indicating the specificity of mutant *GTF2I* in thymomas.

## 3. Discussion

In this study, we investigated the presence of point mutations in *GTF2I* in thymomas using targeted sequencing coupled with molecular barcoding to validate previous findings obtained by whole-exome sequencing [10,11]. We demonstrated a widespread distribution of mutant *GTF2I* in all types of thymomas, including type B. The SIFT and Polyphen 2 algorithms predicted that the *GTF2I* mutation (p.L424H) was somatic and altered protein structure and function [13,14]. Mutant *GTF2I* did not induce anchorage-independent growth but accelerated cell proliferation in vitro [10]. Type A thymomas were reported to harbor this point mutation [10,11]; however, our study also demonstrated the prevalence of this mutation in type B thymomas.

The presence of mutant *GTF2I* in type B thymomas, in addition to type A thymomas, in this study, unlike previous studies, can be attributed to three reasons. First, previous studies used macrodissection on formalin-fixed paraffin-embedded surgical specimens, whereas we performed laser-capture microdissection. Thymomas comprise tumor and normal cells. In particular, type B thymomas consist of a significant proportion of lymphocytes [15,16]. Contamination with normal cells reduces the probability of detecting mutations in tumor cells. We used laser-capture microdissection to select tumor cells to maximize the chances of detecting the point mutation.

Second, previous reports used whole-exome sequencing, whereas we performed deep-targeted sequencing with the molecular-barcoding technique. The sequence coverage was more extensive in our study compared to that in previous studies, and thus the sensitivity and specificity of detecting mutant *GTF2I* were theoretically much higher in our study. Third, we excluded the influence of pseudogenes and manually counted the DNA strands harboring the mutation in the real *GTF2I* gene. The pseudogenes could have potentially biased the measurement, thereby reducing the chance of detecting *GTF2I* since the heterozygous mutations were present in 1 out 6 (17%) of the amplicons. Thus, by eliminating such bias, we increased the chances of detecting mutations in *GTF2I* by approximately six-fold. Meanwhile, Feng et al. demonstrated that the *GTF2I* mutation was detected by quantitative real time PCR and the fraction of mutant *GTF2I* was the highest in type A and AB thymomas, followed by type B1, B2, and B3, consistent with our results [17].

The findings in this study will help in developing molecular *GTF2I*-targeted therapies. Over recent years, targeted drugs have been shown to exert dramatic effects on various carcinomas and have revolutionized cancer treatment [18,19]. For example, first-line therapy can be selected based on the gene mutation profiles of individual tumors, including epidermal growth factor receptor-tyrosine kinase inhibitors for lung cancer, mammalian target of rapamycin inhibitors for renal cell carcinoma, and human epidermal growth factor receptor-2 inhibitors for breast cancer [20,21,22,23]. However, there are no efficacious therapeutic agents for thymomas owing to the lack of knowledge regarding driver mutations. Empiric therapies constitute the currently available treatment strategies for advanced-stage thymoma; the outcomes of these therapies are mostly unsatisfactory. The common driver mutation in *GTF2I* was detected in ~64% of all thymomas; thus, molecular targeted therapy for *GTF2I* may be developed as the primary therapeutic strategy for patients with thymomas in the future.

Immunotherapy has been used in treating various carcinomas. Thus, the *GTF2I* point mutation may serve as a neoantigen for use as a therapeutic target. Treatment strategies using this cancer antigen, such as gene therapy, vaccine therapy, and chimeric antigen receptor T cell therapy coupled with the antibody against PD-1, may be promising for patients in the future [24,25]. In this study, PD-L1 expression and the presence of mutant *GTF2I* were inversely correlated; very few patients were positive for both PD-L1 and mutant *GTF2I*. In addition, blood marker data (serum LDH and PLR) in our study suggested that thymomas without the *GTF2I* mutation exhibited a higher response rate to immunotherapy. We hope that, with further studies, molecular targeted therapy and immunotherapy can be non-redundantly used in different subsets of patients [26].

However, this study is associated with some limitations. First, the patient cohort was relatively small owing to the rarity of the tumor. Second, patient survival could not be analyzed as no patients have shown recurrence in the cohort. Third, we sequenced the cell-free DNA in the serum of all patients, and mutant *GTF2I* could not be detected in these DNA samples. Liquid biopsies utilizing this mutation are deemed unavailable based on our data. In this context, a larger series will be needed to more comprehensively evaluate the genomic landscape of thymomas and more clearly elucidate associations with clinical parameters in a more comprehensive multivariate analysis. However, as the major aim of this preliminary analysis was to identify the driver mutation that should be prioritized for clinical development, the modestly sized sample can still provide useful insights.

## 4. Methods

### 4.1. Patient Cohort and Sample Preparation

In this study, we unbiasedly enrolled 22 patients who underwent surgical resection for thymoma at our hospital between January 2014 and August 2019. We obtained written informed consent for genetic research from all the patients in accordance with the protocols approved by the Institutional Review Board at our hospital (Institutional Review Board at Yamanashi Central Hospital). The specimens were categorized histologically based on the classification guidelines by the World Health Organization [16,27], and staged according to the Masaoka staging system [7,15,28]. Sections of formalin-fixed and paraffin-embedded tissues were stained with hematoxylin-eosin and microdissected using the ArcturusXT laser-capture microdissection system (Thermo Fisher Scientific, Waltham, MA, USA). For type AB thymomas, the type A and B portions were microdissected and examined separately. The GeneRead DNA FFPE Kit (Qiagen, Hilden, Germany) was used according to the manufacturer’s instructions, and the DNA quality was checked using primers against ribonuclease P.

### 4.2. Targeted Deep Sequencing and Data Analysis

There were two pseudogenes with 99.4% sequence homology within approximately 500 base pairs upstream and downstream of the mutation site in *GTF2I* that limited the detection of *GTF2I* mutations. A single base difference (cytosine (C) in *GTF2I* and thymine (T) in the pseudogenes) upstream of the mutation site was used to identify *GTF2I* (Figure 3). Thus, we designed our primers for the polymerase chain reaction of the region inclusive of this single nucleotide variation. The primers were designed for use in targeted sequencing using Ion AmpliSeq Designer (Thermo Fisher Scientific) as described previously [29,30,31,32,33,34,35,36,37,38].

Multiplex PCR was performed with an Ion AmpliSeq HD primer and Ion AmpliSeq HD Library Kit (Thermo Fisher Scientific) in accordance with the manufacturer’s instruction. Primer sets comprised two different primer pools. The reaction mixture comprised 3.7 μL of 4× Amplification Mix, 1.5 μL of 10× forward primer mix, 1.5 μL of 10× reverse primer mix, 1–20 ng of FFPE or plasma DNA, and nuclease-free water up to a 15 μL total volume. PCR was performed to amplify the target regions with the following cycling conditions: three cycles of 99 °C for 30 s, 64 °C for 2 min, 60 °C for 6 min, and 72 °C for 30 s; 72 °C for 2 min; and a final hold at 4 °C. After combining the PCR products, the amplicons were partially digested with 5 μL of SUPA reagent. The reactions were performed using the following conditions: 30 °C for 15 min, 50 °C for 15 min, 55 °C for 15 min, 25 °C for 10 min, 98 °C for 2 min, and a hold at 4 °C. The libraries were amplified with 4 μL of Ion AmpliSeq HD Dual Barcode Kit with the following conditions: 99 °C for 15 s; 5 cycles of 99 °C for 15 s, 62 °C for 20 s, and 72 °C for 20 s; 15–17 cycles of 99 °C for 15 sec and 70 °C for 40 s; 72 °C for 5 min; and a hold at 4 °C.

The sequencing libraries were prepared using the Ion AmpliSeq™ HD Library Kit (Thermo Fisher Scientific) as previously described [39]. After barcoding with Ion AmpliSeq HD Dual Barcode Kit (Thermo Fisher Scientific), the libraries were purified using Agencourt AMPure XP (Beckman Coulter, Brea, CA, USA) and quantified using the Ion Library Quantitation Kit (Thermo Fisher Scientific). Emulsion PCR and chip loading was performed on the Ion Chef with the Ion 540 Kit-Chef or Ion PI Hi-Q Chef kit; sequencing was performed using Ion 540 Kit-Chef on the Ion GeneStudio S5 Prime System or Ion PI Hi-Q Sequencing Kit on an Ion Proton Sequencer (Thermo Fisher Scientific).

### 4.3. Molecular Barcoding

The raw data were analyzed using Torrent Suite version 5.10.0 and processed using the standard Ion Torrent Suite Software running on the Torrent Server. The pipeline consisted of signal processing, base calling, quality score assignment, read alignment to the human genome 19, quality control of the mapping, and coverage analysis. Single nucleotide variants, insertions, and deletions were annotated using the Ion Reporter Server System (Thermo Fisher Scientific). The data were visualized with the Ion Reporter™ Genomic Viewer. We manually counted the *GTF2I* DNA strands with C at the site of the single nucleotide variation. In such DNA strands, the substitution of T>adenine (A) at the hotspot (c.74146970) was considered as a true mutation in *GTF2I* and used for analysis (Figure 3).

### 4.4. PyClone Analysis

PyClone is a Bayesian clustering tool to group sets of deep sequenced somatic mutations into putative clonal clusters while estimating their cellular prevalence. This method accounts for allelic imbalances introduced by changes in the segment copy number and sample contamination by normal cells [40]. In this study, PyClone analysis was performed to estimate the fraction of cancer cells harboring mutant *GTF2I* [40,41,42].

### 4.5. Immunohistochemistry for PD-L1

Specimens from 20 patients obtained between January 2000 and December 2013 were fixed with 10% buffered formalin. Formalin-fixed paraffin-embedded tissues were cut into 5 μm sections, deparaffinized, rehydrated, and stained in an automated system (Ventana Benchmark ULTRA system; Roche, Tucson, AZ, USA) using commercially available detection kits and antibodies against PD-L1 (28–8, ab205921; Abcam, Cambridge, MA, USA). PD-L1 was primarily localized to the cell membrane of tumor cells, and its expression was determined quantitatively by two pathologists based on the proportion of PD-L1-positive tumor cells. Cells were considered PD-L1-positive based on a ≥1% PD-L1 expression.

### 4.6. Presence or Absence of Gtf2i Mutation in Other Malignant Diseases

Other samples, such as brain cancer, lung cancer, gastric cancer, colorectal cancer, hepatocellular carcinoma, breast cancer, and lymphoma, were collected at our institution during regular clinical practice. After obtaining signed informed consent, their sample tissues were analyzed for the presence of the *GTF2I* mutation.

### 4.7. Statistical Analyses

Continuous variables were represented as the mean and standard deviation. Categorical variables were compared using the Chi-square test. Multivariate analyses and calculation of the hazard ratio (HR) and 95.0% confidence interval (CI) were performed using JMP (SAS Institute, Cary, NC, USA). Two-tailed *p* < 0.05 was considered statistically significant.

## 5. Conclusions

A missense mutation in *GTF2I* was detected with high prevalence in and specific to thymomas. This mutation may be a major driver mutation in the tumorigenesis of thymomas and serve as a promising therapeutic candidate to be used in “precision medicine” for patients with thymomas.

## Figures and Tables

**Figure 1 cancers-12-02032-f001:**
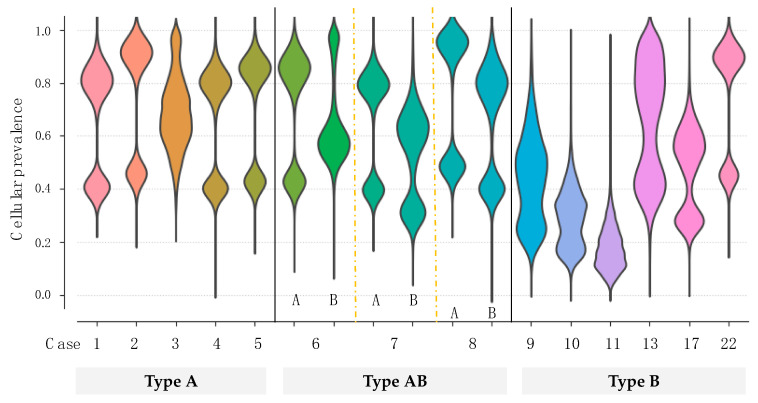
The cellular prevalence of *GTF2I* in the clonal population. The estimated cellular frequency for mutant *GTF2I* is represented by the distribution of the posterior probability using the PyClone model. The colored part represents the distribution of mutant *GTF2I* in each tumor.

**Figure 2 cancers-12-02032-f002:**
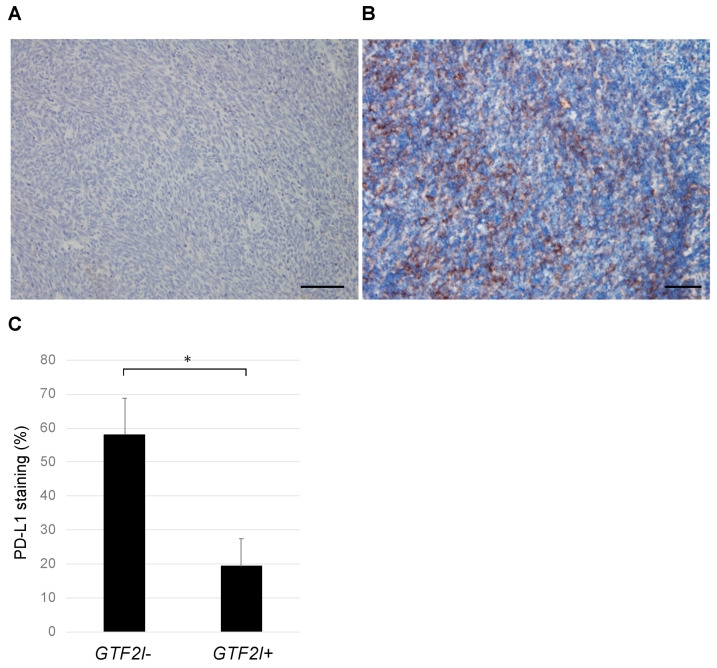
Immunostaining for PD-L1 in thymomas with and without mutant *GTF2I*. (**A**) A representative thymoma with mutant *GTF2I* (Case 1, type A) shows weak staining for PD-L1. (**B**) A representative thymoma without mutant *GTF2I* (Case 15, type B1) shows relatively strong PD-L1 expression. Each scale bar indicates 100 μm. (**C**) The PD-L1 levels were significantly higher in thymomas without mutant *GTF2I*. *, *p* < 0.05.

**Figure 3 cancers-12-02032-f003:**
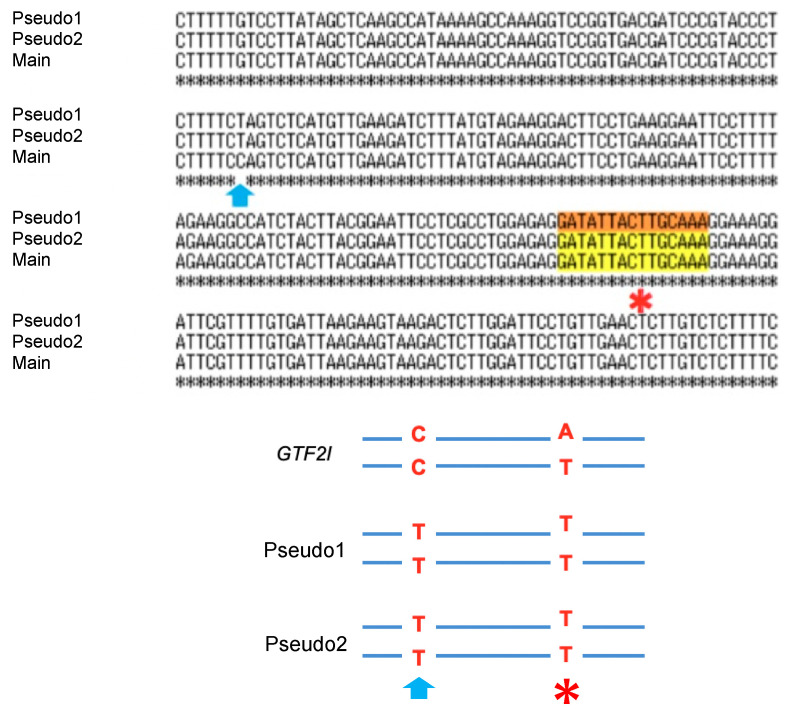
Sequence alignment of *GTF2I* (main) and its pseudogenes showing the single nucleotide variation (as indicated by the arrow) upstream of the point mutation (*). The substitution mutation (* mutation site, thymine (T)>adenine (A)) on the true *GTF2I* DNA strand with cytosine pointed by the arrow was categorized as the true mutation.

**Table 1 cancers-12-02032-t001:** Patient Characteristics.

Parameter		Number of Patients	Overall Percentage
Total number		22	
Age (years), median (range)		66 (42–81)	
Sex			
	Male	12	54.5%
	Female	10	45.5%
Histology			
	Type A	5	22.7%
	Type AB	3	13.6%
	Type B1	7	31.8%
	Type B2	5	22.7%
	Type B3	2	9.1%
Tumor size (cm)			
	≤ 3	9	40.9%
	3 < size ≤ 5	9	40.9%
	5<	4	18.2%
Masaoka Stage			
	I	7	31.8%
	II	12	54.5%
	III	2	9.1%
	IV	1	4.5%
Smoking Status (B.I.) ^a^			
	0	8	36.4%
	1 < B.I. ≤ 600	10	45.5%
	600<	4	18.2%
Myasthenia gravis			
	+	1	4.5%
	−	21	95.5%

^a^ B.I., Brinkman index.

**Table 2 cancers-12-02032-t002:** Characteristics of the Genomic Clusters.

Patient	Age	Sex	Masaoka Stage	Histology	*GTF2I* AF ^b^ (%)	Coverage (Nucleotides)	PD-L1 (%)
1	71	M ^a^	I	A	40.6	1651	1<
2	65	M	I	A	45.7	1793	0
3	80	F ^a^	III	A	66.7	1801	0
4	65	M	I	A	36.3	3401	30
5	68	F	II	A	42.8	3343	80
6	76	M	II	AB-A	34.3	2149	0
				AB-B	11.4	2780	3
7	62	M	I	AB-A	35.8	2675	0
				AB-B	9.4	2342	0
8	45	F	I	AB-A	38.8	7557	0
				AB-B	16.0	6065	10
9	42	F	II	B1	4.5	5639	0
10	76	F	II	B1	5.0	1623	1
11	48	F	II	B1	2.0	1867	0
12	73	M	II	B1	N.D ^c^	−	
13	46	M	II	B1	4.3	5158	7
14	76	F	II	B1	N.D	−	70
15	66	M	II	B1	N.D	−	55
16	76	M	I	B2	N.D	−	70
17	65	M	II	B2	14.1	1652	70
18	53	F	I	B2	N.D	−	50
19	67	M	IV	B2	N.D	−	70
20	44	F	II	B2	N.D	−	60
21	81	M	III	B3	N.D	−	90
22	81	F	II	B3	40.5	9140	80

^a^ M, male; F, female. ^b^ AF, allele fraction. ^c^ N.D, not detected.

**Table 3 cancers-12-02032-t003:** The Presence of Mutant *GTF2I* in Other Cancers.

Type of Malignancy	Age (Mean ± SD)	Sex (Male/Female)	Frequency of Mutant *GTF2I*
Brain cancer	51.0 ± 15.5	12/8	0/20
Lung cancer	69.4 ± 8.6	13/7	0/20
Gastric cancer	71.2 ± 11.5	11/9	0/20
Colorectal cancer	65.8 ± 10.6	13/7	0/20
Hepatocellular carcinoma	68.2 ± 11.0	19/1	0/20
Breast cancer	52.7 ± 12.6	0/20	0/20
Lymphoma	58.9 ± 14.1	12/8	0/20

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
