# Peer review of "Primary Driver Mutations in GTF2I Specific to the Development of Thymomas"

_cancers, 2020, doi:10.3390/cancers12082032_

Round 1

Reviewer 1 Report

The work of Higuchi et al relates the expression of A missense mutation in GTF2I detected with high prevalence in specific thymomas.

General comment:

The data are clear though perhaps a bit too concise. The authors could increase the descriptions of the results obtained by analysis. Also adding final considerations at the end of each paragraph of the results. The work must be expanded to be suitable for publication.

The following questions would serve to clarify some aspects that should be explored or clarified by the authors.

Table 1, from the analysis made there is a lot of variability, the authors should increase the explanations of this.

Figure 1: only two lines of results to describe a complex analysis. It should be explained in more detail

How was the percentage of PDL1 evaluated, was a wester blot of the analyzed tissues or a real-time transcript analysis done? I find the analysis presented insufficient. I would advise the authors to integrate.

Has PDL1 expression also been evaluated in other types of cancer?

How many samples were tested on other types of tumours? Were these also related to gender, age and habits? The authors should integrate this information in section 2.6.

"The buffy coat was separated using centrifugation and DNA was extracted using the QIAamp DNA 182 Blood Mini Kit (Qiagen); this served as the control.

I'm not clear how the PBMCs were used. How do you control what? Expression? The results are inconclusive.

Has circulating PDL1 been evaluated?

Are there any correlations even with clinical parameters of the patients? With the inflammatory state, with the immune system? Considering the importance of the immune system in immunotherapy, these aspects should be analysed and correlated in the cohort of patients with PDL1 and GTF2I expression.

Author Response

Comment 1

The data are clear though perhaps a bit too concise. The authors could increase the descriptions of the results obtained by analysis. Also adding final considerations at the end of each paragraph of the results. The work must be expanded to be suitable for publication.

Response: According to the reviewer’s suggestion, we added several explanatory descriptions to each subsection in the Results section.

Comment 2

Table 1, from the analysis made there is a lot of variability, the authors should increase the explanations of this.

Response:

According to the reviewer’s suggestion, we added the description to section 2.1., as follows.

“The 22 patients recruited in this study were divided according to the Masaoka stages: stage I (n = 7), II (n = 12), III (n = 2), and IV (n = 1). The maximum tumor diameter ranged from 20 mm to 95 mm (mean tumor diameter, 43.6 ± 22.8 mm).”

Comment 3

Figure 1: only two lines of results to describe a complex analysis. It should be explained in more detail

Response:

According to the reviewer’s suggestion, we added these descriptions to section 2.3.

“In our analysis of somatic mutations, there was a need to alleviate the allelic imbalances due to normal-cell contamination, especially in lymphocyte-rich type B thymomas. In this context, PyClone analysis was performed to estimate the cellular frequency patterns of mutations in a population of tumor cells.”

“This GTF2I mutation appeared to trigger clonal expansion and is retained ubiquitously within the tumors of the same clone.”

Comment 4

How was the percentage of PDL1 evaluated, was a wester blot of the analyzed tissues or a real-time transcript analysis done? I find the analysis presented insufficient. I would advise the authors to integrate.

Response: According to the reviewer’s suggestion, we revised the phrase in section 2.4.

“analyzed”→ “evaluated immunohistochemically”

Immunohistochemical evaluation is described in section 4.5.

Comment 5

Has PDL1 expression also been evaluated in other types of cancer?

Response: The key point of our study is that GTF2I mutation can be detected in thymomas in general and it may serve as a novel therapeutic target. We feel that the examination of PD-L1 in other cancer types is beyond the scope of our current study.

Comment 6

How many samples were tested on other types of tumours? Were these also related to gender, age and habits? The authors should integrate this information in section 2.6.

Response: According to the reviewer’s suggestion, we added some information in section 2.7.

Comment 7

"The buffy coat was separated using centrifugation and DNA was extracted using the QIAamp DNA 182 Blood Mini Kit (Qiagen); this served as the control.

I'm not clear how the PBMCs were used. How do you control what? Expression? The results are inconclusive.

Response: Thank you for your well-directed suggestion. Actually, we did not use buffy coat as a control in the analyses. We deleted theese sentences.

Comment 8

Has circulating PDL1 been evaluated?

Response: We did not check circulating PD-L1 in this study, however, it seems quite an interesting topic, and hopefully may be our next research theme. Thank you for your kind suggestion.

Comment 9

Are there any correlations even with clinical parameters of the patients? With the inflammatory state, with the immune system? Considering the importance of the immune system in immunotherapy, these aspects should be analysed and correlated in the cohort of patients with PDL1 and GTF2I expression.

Response: According to the reviewer’s suggestion, we analyzed the correlation between GTF2I/PD-L1 and peripheral blood parameters. We added these descriptions in the Results and Discussion sections, as follows.   

“Peripheral blood markers were examined to better characterize the cases in reference to the clinical benefits in immunotherapy. An elevated lactate dehydrogenase (LDH) level was reported to be indicator of tumor burden that is typically associated with lower response rates to immunotherapy, while an elevated platelet-lymphocyte ratio (PLR) was also associated with lower response rates in patients treated with immunotherapy [12]. The pre-surgery blood exam data were analyzed, which revealed that serum LDH levels were significantly higher in thymomas with mutant GTF2I than those with wildtype GTF2I (thymomas with mutant GTF2I, 224.3±13.3; thymomas with wildtype GTF2I, 183.7±12.1; p<0.05). In addition, the PLR was significantly higher in thymomas with mutant GTF2I than those with wildtype GTF2I (thymomas with mutant GTF2I, 137.2±13.1; thymomas with wildtype GTF2I, 97.0±15.7; p<0.05).”

“Blood marker data (serum LDH and PLR) in our study suggested that thymomas without the GTF2I mutation exhibited a higher response rate to immunotherapy.”

I believe your comments have improved our manuscript so much. Thank you very much for your thoughtful comments.

Reviewer 2 Report

This article features the analysis of 22 surgical specimen of thymoma (21 resectional surgery and 1 biopsy) using laser-capture microdissection to avoid contamination of normal cells, targeted sequencing with molecular barcoding techniques to detect mutant GTF2I, and PyClone analysis to determine the fraction of tumor cells harboring mutant GTF2I. While mutant GTF2I was not detected in malignancies such as lung, gastric, colorectal and liver cancer, the authors demonstrated high cellular prevalence of mutant GTF2I in thymoma tumor cells.

The association of GTF2I mutation with the pathogenesis of thymoma has been reported previously using different genomic platform. The fraction of mutant GTF2I was documented to be the highest in type A thymoma, followed by type AB, B1, B2, and B3. Also, thymomas bearing GTF2I mutations had a better prognosis than those bearing wild-type GTF2I, coincident with the results of this study.

The manuscript was well written and sound in methodology, confirming decreased fraction of mutant GTF2I in type B thymomas compared to type A and AB thymomas. However, the Masaoka stage of each tumor would be best listed in Table 2 so that the readers could have a better understanding.

Author Response

The association of GTF2I mutation with the pathogenesis of thymoma has been reported previously using different genomic platform. The fraction of mutant GTF2I was documented to be the highest in type A thymoma, followed by type AB, B1, B2, and B3. Also, thymomas bearing GTF2I mutations had a better prognosis than those bearing wild-type GTF2I, coincident with the results of this study.

Response: According to the reviewer’s suggestion, we added the description in Discussion section, as follows.

“Meanwhile, Feng et al. demonstrated that the GTF2I mutation was detected by quantitative real time PCR and the fraction of mutant GTF2I was the highest in type A and AB thymomas, followed by type B1, B2, and B3, consistent with our results.”

The manuscript was well written and sound in methodology, confirming decreased fraction of mutant GTF2I in type B thymomas compared to type A and AB thymomas. However, the Masaoka stage of each tumor would be best listed in Table 2 so that the readers could have a better understanding.

Response: According to the reviewer’s suggestion, we added the information of Masaoka stage to Table 2.

Thank you very much for your thoughtful comments.

Reviewer 3 Report

In the manuscript, the authors present a study regarding the incidence of the GTF2I mutations in patients with thymomas. They introduced in the study, 22 subjects with type B thymomas. In my opinion it is an interesting manuscript, that can be published. Also, in order to improve the quality of the manuscript some changes have to be done. My observations are :

  1. Regarding the incidence of GTF2I mutations in other types of cancers, you have to introduce in the manuscript some data regarding the patients that were included in the study.
  2. The authors did not specified if they introduced in the study patients with micronodular thymomas. 

Author Response

  1. Regarding the incidence of GTF2I mutations in other types of cancers, you have to introduce in the manuscript some data regarding the patients that were included in the study.

Response: According to the reviewer’s suggestion, we added the description in section 2.7., as follows.

“In order to examine the specificity of the GTF2I mutation, patients with other malignant diseases were also enrolled in the study. The patients who underwent surgery or biopsy at our hospital between January 2014 and August 2019 were enrolled without bias, and they exhibited a wide range of histology and stages.”

We also added the data of age and gender to Table 3.

  1. The authors did not specified if they introduced in the study patients with micronodular thymomas. 

Response: There were no micronodular thymomas in our study. We added this information to section 2.1.

Thank you very much for your thoughtful comments.

Reviewer 4 Report

Rumi Higuchi et al motivate their paper on the importance of mutations in thymic epithelial tumours of which I agree with. It is true type A and AB thymoma present mostly with low malignant potential, whereas type B1, B2, and B3 thymoma are more aggressive, with B3 thymoma having the greatest tendency for mostly intrathoracic spread. Thus, focusing on the importance of mutations in GTF2I in the development of type B thymomas using targeted sequencing coupled with sensitive and specific assays to create novel therapeutic strategy is a good idea. Overall, the paper is well written and brings out clear justifications of the objective.

Minor points:

The authors should consider revising the manuscript to remove minor editorial errors such as spaces, spelling, and the clarity in sentence construction.

The authors should consider revising the manuscript to so that there is consistent referencing without using website links.

The authors should consider revising the references so that where there are more than five authors, then a standard format should be adhered to.

Author Response

 Minor points:

Comment 1

The authors should consider revising the manuscript to remove minor editorial errors such as spaces, spelling, and the clarity in sentence construction.

Response: We had all the manuscript checked and revised by the MDPI editing service.

Comment 2

The authors should consider revising the manuscript to so that there is consistent referencing without using website links. The authors should consider revising the references so that where there are more than five authors, then a standard format should be adhered to.

Response: We prepared the references with a software Endnote X9. We confirmed the journal’s format, and did our best to comply with it.

Thank you for your thoughtful comments.
